# Green Approach for Synthesizing Copper-Containing ZIFs as Efficient Catalysts for Click Chemistry

**Alireza Pourvahabi Anbari** [1,2], **Shima Rahmdel Delcheh** [3], **Philippe M. Heynderickx** [2,4], **Somboon Chaemcheun** [5], **Serge Zhuiykov** [2,6] and **Francis Verpoort** [5,7,*]

1 Department of Chemistry, Faculty of Science, Ghent University, 9000 Ghent, Belgium; alireza.pourvahabi@ghent.ac.kr
2 Center for Environmental and Energy Research (CEER), Ghent University Global Campus, Incheon 406-840, Republic of Korea; philippe.heynderickx@ghent.ac.kr (P.M.H.); serge.zhuiykov@ghent.ac.kr (S.Z.)
3 Department of Chemistry, Guilan University, Rasht P.O. Box 1841, Iran; shima.rahmdel@yahoo.com
4 Department of Green Chemistry and Technology, Faculty of Bioscience Engineering, Ghent University, Coupure Links 653, 9000 Ghent, Belgium
5 State Key Laboratory of Advanced Technology for Materials Synthesis and Processing, Wuhan University of Technology, Wuhan 430070, China; sama_che@hotmail.com
6 Department of Solid-State Sciences, Faculty of Science, Ghent University, 9000 Ghent, Belgium
7 National Research Tomsk Polytechnic University, Lenin Avenue 30, 634050 Tomsk, Russia
* Correspondence: francis@whut.edu.cn

**Abstract:** ZIF-8 and ZIF-67 containing various percentages of copper were successfully synthesized through a green in-situ thermal (IST) approach based on 2-methylimidazole (2-MIM) as the organic linker. The IST method has several advantages over previously reported studies, including solvent and additive-free reaction conditions, a mild reaction temperature, a single-step procedure, no activation requirements, and the use of the smallest precursor ratio (M/L). The high catalytic performance of Cu/ZIF-8 and Cu/ZIF-67 in click chemistry is attributed to their high specific surface area, excellent porosity, and structural stability. To achieve these features, a range of parameters—such as time, temperature, gas atmosphere, and precursor ratio—were optimized. Several characterization methods were used to confirm the features of the produced catalysts. Overall, the synthesis strategy for achieving the targeted ZIFs with unique features is "green" and does not require further activation or treatment to eliminate side products. This method has great potential for manufacturing metal-organic frameworks on a large scale. Moreover, water was used as a solvent during the click reaction, resulting in high yields and making this an attractive, green, and eco-friendly procedure.

**Keywords:** zeolitic imidazole frameworks; green chemistry; click reaction; azide-alkyne cycloaddition; CuAAC

## 1. Introduction

Metal-organic frameworks (MOFs) are highly crystalline and porous structures containing metal nodes and organic linkers in an unlimited network. The synthesis and applications of MOFs are still attractive research fields for scientists. Due to their fantastic properties, such as their tunable pore size, the flexibility in their structures, their pore volume, high specific surface area, high stability, etc., MOFs could be unique candidates for use in a vast range of applications such as storage, sensors, catalysis, drug delivery, etc. [1–5]. Zeolitic-imidazole frameworks (ZIFs), a subgroup of MOFs containing metal centers in coordination with imidazole's, can construct a tetrahedral structure with a so-dalite topology. A wide range of metals such as Zn, Co, Mn, Fe, and Ni can take part in the synthesis of ZIFs. The organic linkers 2-nitroimidazole (NIM), 1-ethylimidazole (EIM), 2-methylimidazole (2-MIM), and 1-methylimidazole (MIM) are mostly used for synthesizing ZIFs [6]. The properties of ZIFs are completely defined by the nature of the

metal, organic linkers, and reaction conditions. Due to some of their interesting properties, such as their high porosity and extra thermal stability, ZIFs could be better candidates than activated carbon, zeolites, and metal oxides for use in applications such as catalysis, biofuel production, drug delivery, and separation—as demonstrated by many reports [6–8].

There are many methods for synthesizing MOFs and ZIFs for various applications. Generally, in traditional procedures (solvothermal), the metal source and organic linkers are dissolved in an organic solvent under different reaction conditions. However, there are disadvantages, such as the long reaction time, the use of organic solvents (THF, DMF, DEF), and limitations in using precursors and water as green solvents. Additionally, the final obtained ZIFs need further treatment to remove the remaining solvent molecules to improve their suitability for the desired application [9–11]. Some advanced synthesizing methods—such as mechanochemical, microwave-assisted, electrochemical, etc.—are green but pose some disadvantages, such as the need for unique instruments with complicated operational procedures, the use of organic solvents and a lot of energy, prolonged reaction times, and the utilization of some additives. These drawbacks result in limitations in the synthesis and utilization of ZIFs for novel applications. As such, discovering a green, novel, and simple method for synthesizing ZIFs on a large scale will be a challenge for scientists in the future [12–14].

In 1999, Click Chemistry was introduced by Barry Sharples and his group at the annual meeting of the American Chemical Society. In 2001, Sharples described click chemistry as a group of reactions with unique features, such as being very purposeful, having high product yields, and having no by-products produced during the reaction. If by-products are obtained in the reaction, they can be removed from the reaction by non-chromatographic methods. The conditions for the reaction must be straightforward—in other words, the ideal conditions—and the steps during the reaction should be insensitive to oxygen and water. The use of any solvent for the reaction should be avoided, but if a solvent is necessary, a safe and green solvent such as water and ethanol that is easily removable from the reaction should be applied. The separation of products should be straightforward, and if purification is required, non-chromatographic methods such as crystallization or distillation should be used. Additionally, the obtained products must be stable under physiological conditions, and the reaction stages must proceed quickly towards completion so that the reaction has a high selectivity for producing a single unique product [15–17].

Click chemistry is divided into four categories: cycloaddition reactions [18], ring-opening reactions of heterocycles [19], carbonyl chemistry of non-aldol species [20], and addition reactions to multiple C–C bonds [21,22]. Nowadays, click chemistry stands as a powerful concept for reaching different types of triazoles that are often used in pharmaceutical materials, drug manufacturing, polymers, and materials science. The Huisgen cycloaddition process consists of the 1,3-dipolar cycloaddition of a terminal alkyne and organic azide, catalyzed by a copper catalyst to produce 1,2,3-triazoles—the finest and most prominent example of click chemistry to date [23–25]. These 1,2,3-Triazoles have antimicrobial, antibacterial, and antiviral properties, and research is ongoing to find new biologically active properties of 5,2,3-triazoles. Recently, 1,2,3-triazoles—considering their applications in pharmaceutical and agricultural industries—have been highly demanded. In addition to medicinal applications where biological activity is very important, these triazoles are also used in materials science and biochemistry [26–36].

Copper-catalyzed azide-alkyne cycloaddition (CuAAC), as a subgroup of click chemistry, is a most attractive and fundamental ligation reaction in nature between organic azides and terminal alkynes that produces 1,2,3-triazole derivatives. These 1,2,3-triazole compounds have been extensively useful in a diversity of chemical fields, such as the modification of biological macromolecules, organic synthesis, and polymer and materials chemistry. Recently, the use of ZIFs as heterogeneous catalysts has been increasing dramatically [37–39]. At present, metal-organic frameworks—specifically zeolitic imidazole frameworks—exhibit great potential for several applications due to their specific properties, being environmentally friendly and cost-effective synthesis methods. ZIFs, having an abun-

dance of active sites—both basic (imidazole) and acidic (metal nodes) sites—have become affordable materials for different types of catalytic reactions. Enhancing the crystallization rate is crucial for generating uncoordinated sites (defect structures), and thus catalytic activity. Compared with traditional synthesis methods, fast crystallization through the IST method (solvent-free) results in several defect structures. Both $Cu_{10}ZIF-8$ and $Cu_{20}ZIF-67$, obtained through the IST method, exhibit high catalytic activity and selectivity under mild reaction conditions without the presence of a co-catalyst. Additionally, during CuAAC, high thermal and chemical stability (robustness) has been demonstrated by the reusability of both $Cu_{10}ZIF-8$ and $Cu_{20}ZIF-67$ as heterogeneous catalysts. These catalysts can be reused four times without a significant decrease in catalytic performance. After four cycles, XRD and SEM analysis were applied to determine the structural integrity. Herein, we report an eco-friendly and simple synthesis approach (IST method) for $Cu_{10}/ZIF-8$ and $Cu_{20}/ZIF-67$ containing different percentages of copper and used as efficient heterogeneous catalysts for Azide-Alkyne cycloaddition. Reaction parameters such as contact time, temperature, catalyst dosage, solvent, and substrate were optimized. The catalysts demonstrated a high catalytic activity in water as a green solvent and could be easily separated from the reaction mixture by centrifugation. Finally, XRD, FTIR, EDS, TEM, BET, TGA, and ICP-OES were used to investigate the physical, chemical, and structural features of the ZIFs. The outcomes of this research included: (1) the modification of the ZIF-8 and ZIF-67 structures with different amounts of copper through the IST method; (2) the investigation of their catalytic performance in Azide-Alkyne cycloaddition in water; and (3) an analysis of their different catalytic parameters.

## 2. Results

Up to now, a high range of ZIFs have been synthesized, of which, zeolitic imidazole frameworks based on 2-MIM linkers are applied in a vast range of applications—specifically ZIF-8, ZIF-67, and their derivatives exhibit impressive properties and performances in various fields such as catalysis, separation, adsorption, etc. [40–42]. To obtain the optimized characteristics of the targeted ZIFs, the main parameters—the ratio of nodes/linkers, number of precursors, temperature, and time—were adjusted. Combinations of zinc acetylacetonate hydrate, copper (II) acetylacetonate, and cobalt (II) acetylacetonate, with 2-methylimidazole as an organic linker, were heated in a tubular furnace to synthesize Cu-ZIF-8 and Cu-ZIF-67 under an inert atmosphere to prevent the risk of contamination from reactive gases that exist in the air, such as oxygen and water vapor. High levels of porosity, crystallinity, and specific surface area with high stability are the common properties of zeolitic imidazole frameworks. To verify these characteristics, analytic techniques such as XRD, BET, and TGA can be applied. Throughout the ZIFs' synthesis procedure, the mole ratio of metal nodes/organic linkers is the most critical factor [43,44].

To achieve a better understanding of their crystalline properties, samples of ZIF-8 and ZIF-67 with different amounts of copper were investigated by powder X-ray diffraction (XRD), as shown in Figure 1. The XRD pattern revealed that Cu-ZIF-8 and Cu-ZIF-67 showed a body-centered cubic and polyhedral crystal structure, respectively—which is in excellent agreement with the reported data [45,46].

By increasing the percentage of copper in both ZIF-8 and ZIF-67, a decline was observed in the intensity of the first diffraction peak. The absence of any other peaks in the XRD patterns reveals three important facts; firstly, crystal formation between the 2-MIM and the metal (Cu, Co, Zn) in solvent-free conditions occurs under thermal treatment. Secondly, Cu/ZIF-8 and Cu/ZIF-67 were successfully synthesized without any damage to the framework. Thirdly, the synthesized products showed a high level of purity. These outcomes reveal that the IST procedure for gaining Cu-ZIF-67 and Cu-ZIF-8 with different amounts of copper (10%, 20%, 30%, 40%, and 50%) was successful. The intensity decrease of the first diffraction peak could also be clarified by the different molar ratios of copper applied during the synthesis; this copper amount might limit the cobalt bridging with 2-MIM in the framework compared to the theoretical ratio. The deficiency of the coordina-

tion between cobalt and 2-MIM in the case of the Cu-ZIF-67 was also reflected by the color difference of the materials; the materials with a higher ratio were deep purple to navy blue colored (Figure S4).

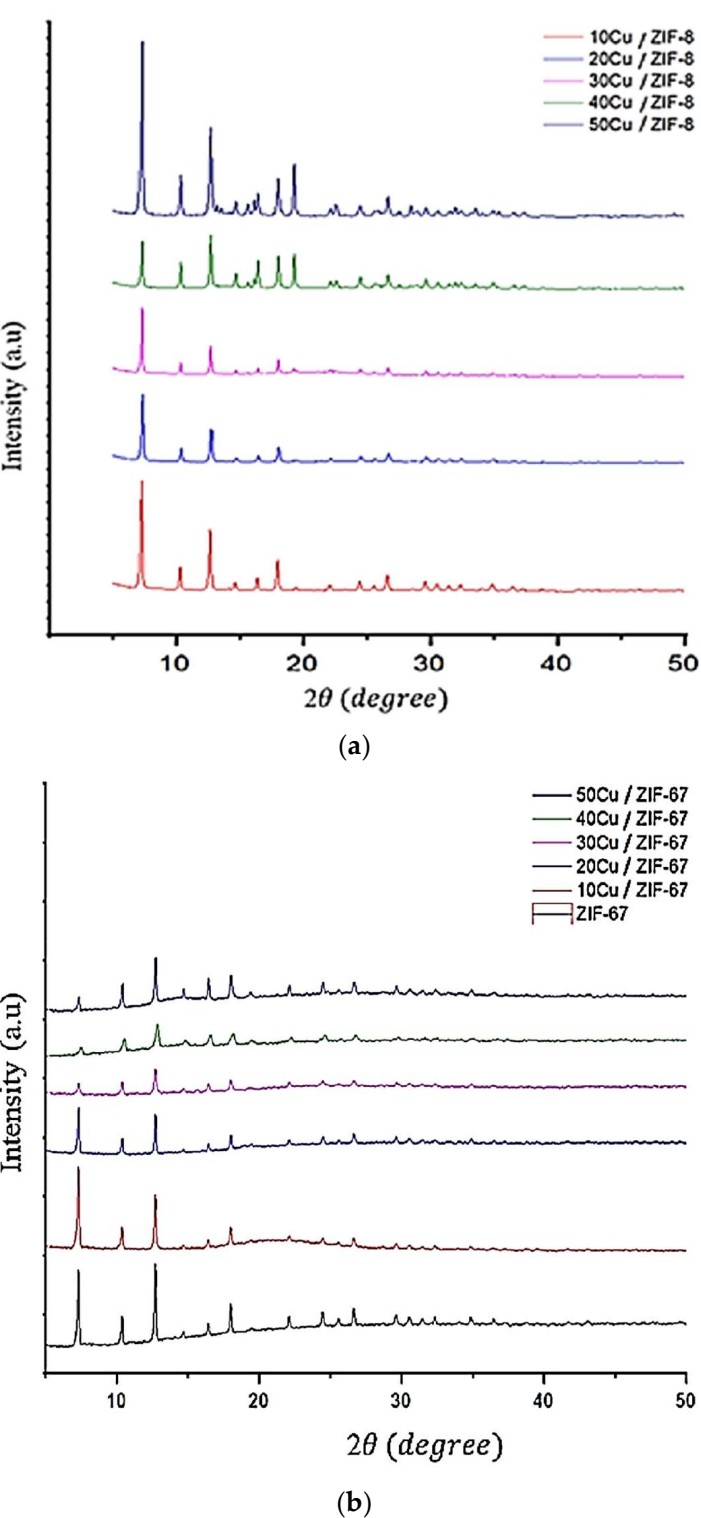

(a)

(b)

**Figure 1.** XRD patterns (**a**) Cu/ZIF-8 with different amounts of Copper (**b**) Cu/ZIF-67 with different amounts of Copper. (IST method: thermal treatment at 200 °C for 1 h under a $N_2$ atmosphere (200 cm$^3$ min$^{-1}$).

The permanent porosity, microporous structure, and stability of the synthesized samples (Cu$_x$-ZIF-67 and Cu$_x$-ZIF-8) were determined by N$_2$ adsorption. As shown in Figure S5, the samples revealed a nitrogen isotherm type (I). Surprisingly, in the N$_2$ isotherm pattern for the Cu$_{30}$ZIF-67, Cu$_{40}$ZIF-67, and Cu$_{50}$ZIF-67, hysteresis occurred due to differences in pore sizes (presence of mesopores with capillary condensation); this is related to the amount of copper that is incorporated in the framework structure. By increasing the copper amount in the ZIFs' structure, the hysteresis phenomenon occurred when the amount was 30 wt% or higher. Below 30 wt% of copper incorporation in the framework, the hysteresis loop of N$_2$ was absent [44]. In addition, during the IST process, the gas released throughout the chemical reaction, e.g., precursor degradation, can produce hierarchal materials, and thus produce hysteresis in the nitrogen isotherm. In the Cu-ZIF-8 and Cu-ZIF-67 isotherm patterns (up to 20% of Cu-loading), the crystal properties were only a little affected by the Cu$^{2+}$ exchange. The amount of adsorbed nitrogen at −196 °C was used to analyze the surface area and porosity of the samples, see Figure 2.

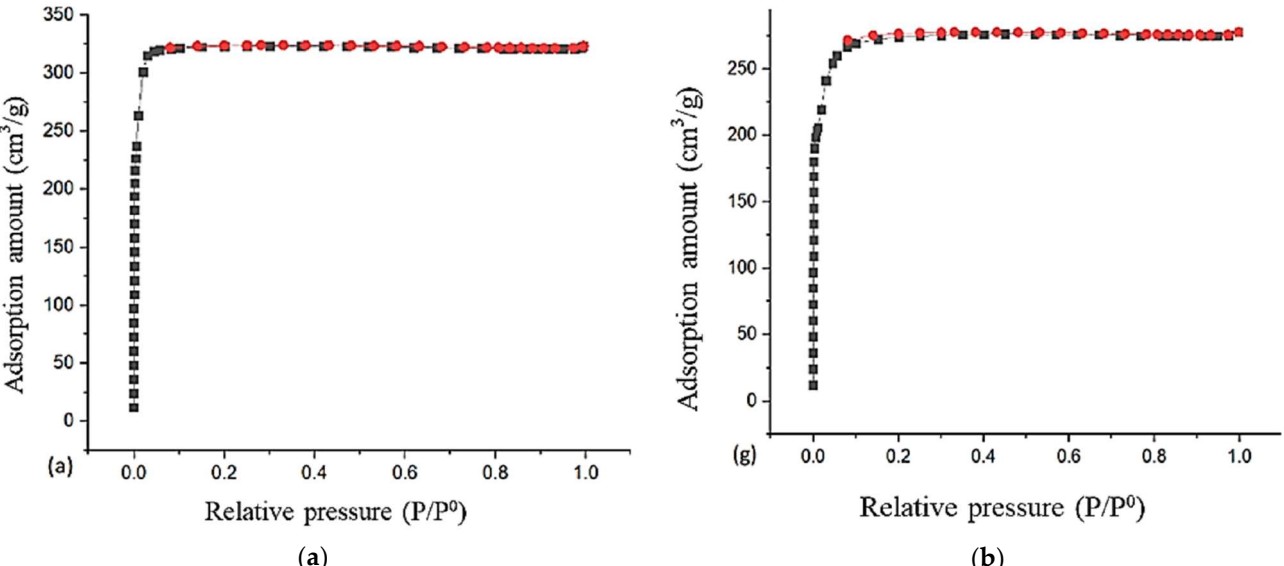

**Figure 2.** N$_2$ adsorption/desorption curves (Black = adsorption; red = desorption) for Cu-ZIF-8 and Cu-ZIF-67 materials with different amount of copper: (**a**) Cu$_{10}$ZIF-8 (**b**) Cu$_{20}$ZIF-67.

As Shown in Tables 1 and S2, the Brunauer–Emmett–Teller method (BET) was applied to determine porosity properties, such as surface area, pore size, pore volume, and the Langmuir surface area of the synthesized ZIFs. With a higher amount of copper in the framework, the surface area and porosity properties decreased in ZIF-8 and ZIF-67 due to impurities that can cause structural changes and even the opening/closing of some pores. Moreover, the porosity properties—such as pore volume and pore size (calculated by the Horvath–Kawazoe method)—displayed a similar behavior as the surface area. The reduction of the molar ratio of 2-MIM to zinc and cobalt resulted in a lower BET surface area and total pore volume of the particles. This reduction was caused by the bigger particle size distributions at the smaller 2-MIM/Zn molar ratio, which confirmed that smaller particles have larger BET surface areas [47].

**Table 1.** Porosity and surface area of Cu$_{10}$ZIF-8 and Cu$_{20}$ZIF-67 samples.

| Sample | BET | Langmuir | Pore Size (nm) | Pore Volume (cm$^3 \cdot$g$^{-1}$) |
|---|---|---|---|---|
| Cu$_{10}$ZIF-8 | 1390.37 | 1477.82 | 1.437 | 0.499 |
| Cu$_{20}$ZIF-67 | 1152.96 | 1256.79 | 1.490 | 0.429 |

To determine the chemical and thermal stability of the $Cu_x$ZIFs, thermogravimetric analysis (TGA) was applied. As shown in Figure 3, weight changes in the copper ZIFs at different temperatures were observed under an $N_2$ atmosphere with a flow rate of 5 cc/min. The TGA pattern for $Cu_{10}$ZIF-8 exhibited two weight losses at 197 °C (17%) and at 600 °C (40%). The TGA pattern for $Cu_{20}$ZIF-67 revealed two weight changes at 353 °C (4.5%) and 560 °C (69.4%). The first weight loss for $Cu_{10}$ZIF-8 was not associated with structural damage but could be related to the removal of adsorbed water or unreacted 2-MIM, and the second weight loss was connected to the degradation or decomposition of the structure of the ZIF-8 and ZIF-67 [48].

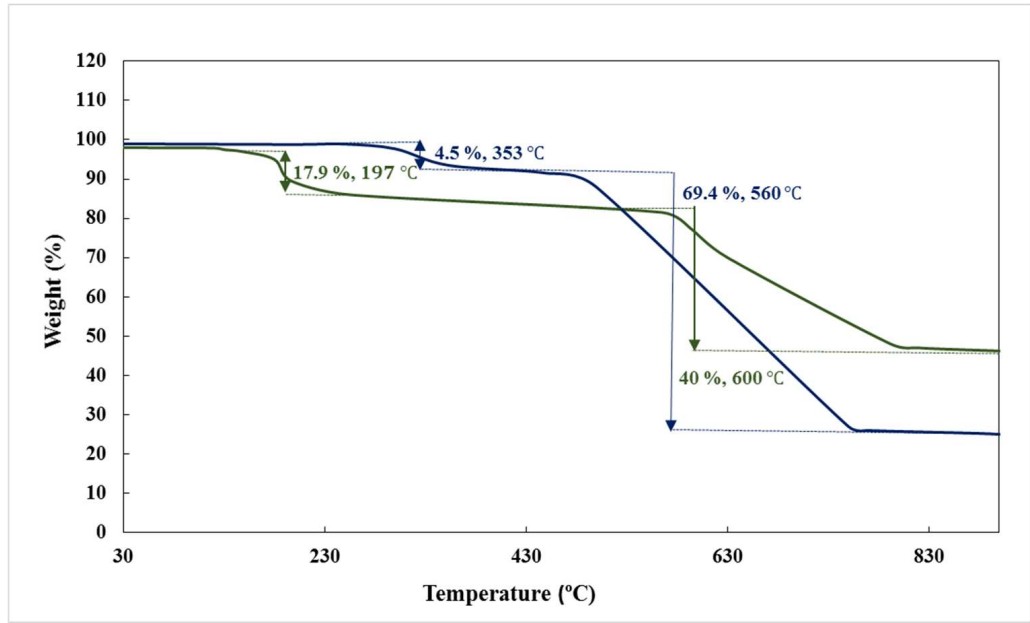

**Figure 3.** TGA pattern of $Cu_{10}$-ZIF-8 (Green) and $Cu_{20}$-ZIF-67 (blue).

As shown in Table 2, inductively-coupled plasma-optical emission spectrometry (ICP-OES) was applied to measure the metallic quantity (Zn, Co, Cu) of the ZIFs. Additionally, ICP-OES was used to determine the amount of copper in $Cu_{20}$ZIF-67 and $Cu_{10}$ZIF-8 after digestion in an $HNO_3$ solution. According to the ICP-OES results, the amount of copper in $Cu_{20}$ZIF-67 and $Cu_{10}$ZIF-8 was 6.34 and 3.74, respectively.

**Table 2.** The copper content from ICP analysis.

| Samples | Co or Zn (%wt) | Cu (%wt) |
|---|---|---|
| $Cu_{20}$ZIF-67 | 22.38 | 6.34 |
| $Cu_{10}$ZIF-8 | 37.80 | 3.74 |

Energy dispersive spectrometry (EDS) mapping was applied to investigate the topological properties and the distribution of elements. Figures 4 and S9 revealed that for both catalysts ($Cu_{10}$ZIF-8 and $Cu_{20}$ZIF-67), Cu, C, N, Zn, and Co were distributed homogeneously. The mapping of Zn, Co, C, N was ascribed to ZIF-8 and ZIF-67. The uniform distribution of Cu confirmed that $Cu_{10}$ZIF-8 and $Cu_{20}$ZIF-67 with an ultrahigh catalytic performance for CuAAC were successfully prepared. Additionally, to determine the elemental composition and molar ratio of the catalysts, energy dispersive X-ray analysis (EDX) was used. The EDX showed that $Cu_{10}$ZIF-8 and $Cu_{20}$ZIF-67 were synthesized with their specific molar ratio and atomic percentage.

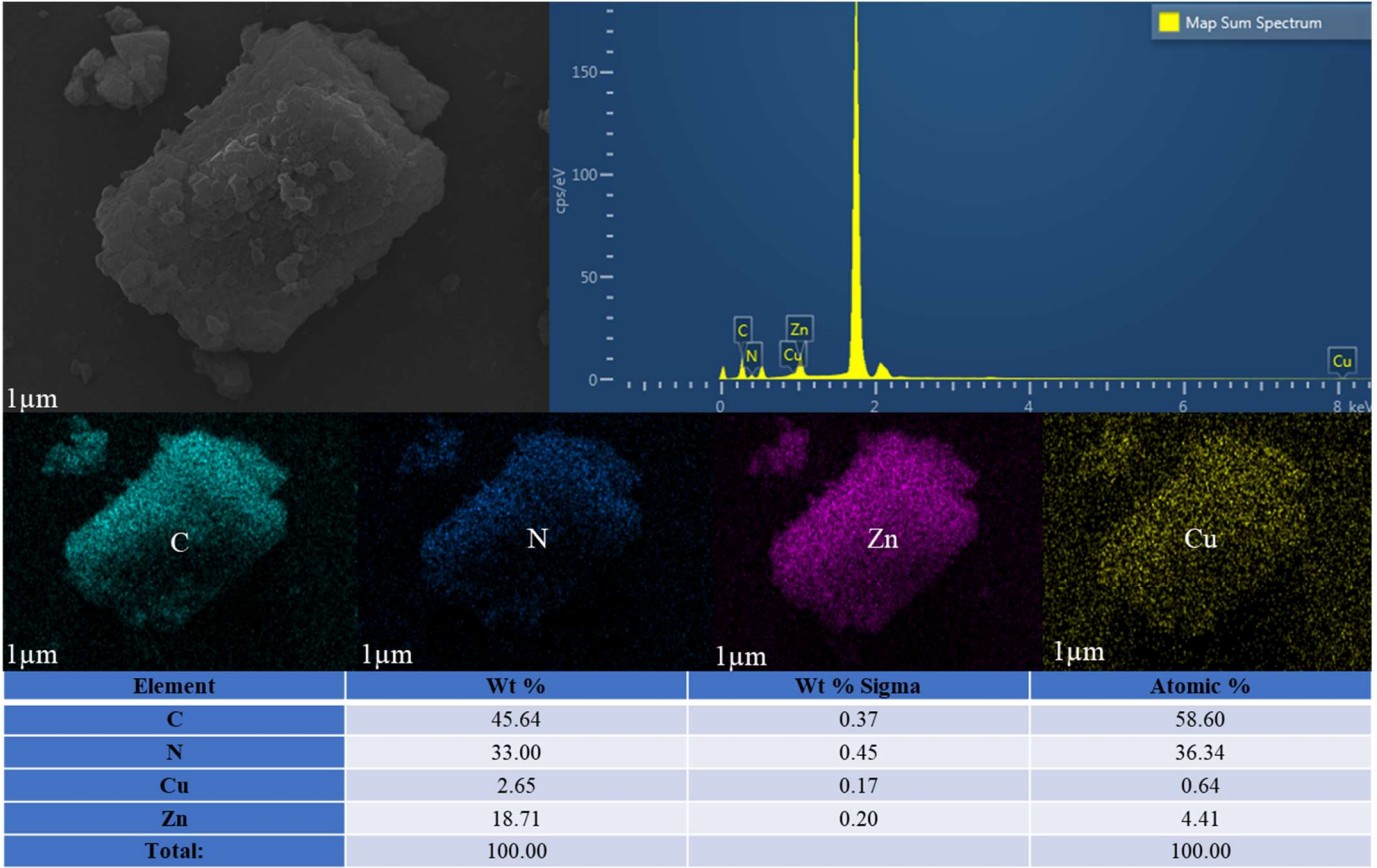

| Element | Wt % | Wt % Sigma | Atomic % |
|---------|------|------------|----------|
| C | 45.64 | 0.37 | 58.60 |
| N | 33.00 | 0.45 | 36.34 |
| Cu | 2.65 | 0.17 | 0.64 |
| Zn | 18.71 | 0.20 | 4.41 |
| Total: | 100.00 | | 100.00 |

**Figure 4.** Energy dispersive spectrometry (EDS) mapping of $Cu_{10}ZIF$-8 obtained by the IST approach.

To find out more about the morphological properties (shape, particle distribution) of $Cu_{10}ZIF$-8 and $Cu_{20}ZIF$-67, scanning electron microscope (SEM) analysis was applied—Figure 5. A rhombic dodecahedron structure (typical ZIF-8 morphology) for $Cu_{10}ZIF$-8 (Figure 5a,b) and a regular rhombic dodecahedral for $Cu_{20}ZIF$-67 (Figure 5c,d) was obtained. Both catalysts had a high distribution of copper and high porosity. In addition, a crystal-like agglomeration into large particles was observed; due to the absence of the solvent in the IST method, a high level of uniform materials could not be achieved. However, obtaining non-uniform particles by applying the IST approach agrees with previous reports in the literature [49,50]. Additionally, transmission electron microscopy (TEM) was used to analyze the catalysts' structural and chemical properties. Random particle sizes from 60 nm up to 600 nm were observed for $Cu_{10}ZIF$-8 and $Cu_{20}ZIF$-67. In addition, TEM confirmed tha the shapes of the $Cu_{10}ZIF$-8 and $Cu_{20}ZIF$-67 were rhombic dodecahedral and regular rhombic dodecahedral, respectively.

Table 3 provides the optimized reaction parameters applied during the azide-alkyne cycloaddition (time, temperature, solvent, amount of catalyst, and different types of organic halides and azides). To confirm the 1,2,3-triazoles' formation, the final products were characterized by $^1$H NMR and $^{13}$C NMR—Figures S3–S6. The obtained results presented in the Supplementary Information revealed that, depending on the Cu percentage inside the ZIFs, the formation of 1,2,3-triazoles (the 1,4 isomer) was complete. ZIF-8 and ZIF-67, without copper in their frameworks, were totally ineffective in the azide-alkyne cycloaddition reaction under the same conditions. Additionally, various organic halides and alkynes were applied in the azide-alkyne cycloaddition catalyzed by the $Cu_{10}ZIF$-8 and $Cu_{20}ZIF$-67, demonstrating the substrate scope of the catalysts—Table S1. It is worth mentioning that this is the first study synthesizing ZIFs containing copper with different

percentages in a short process time without using expensive solvents, high pressure, or different types of precursors.

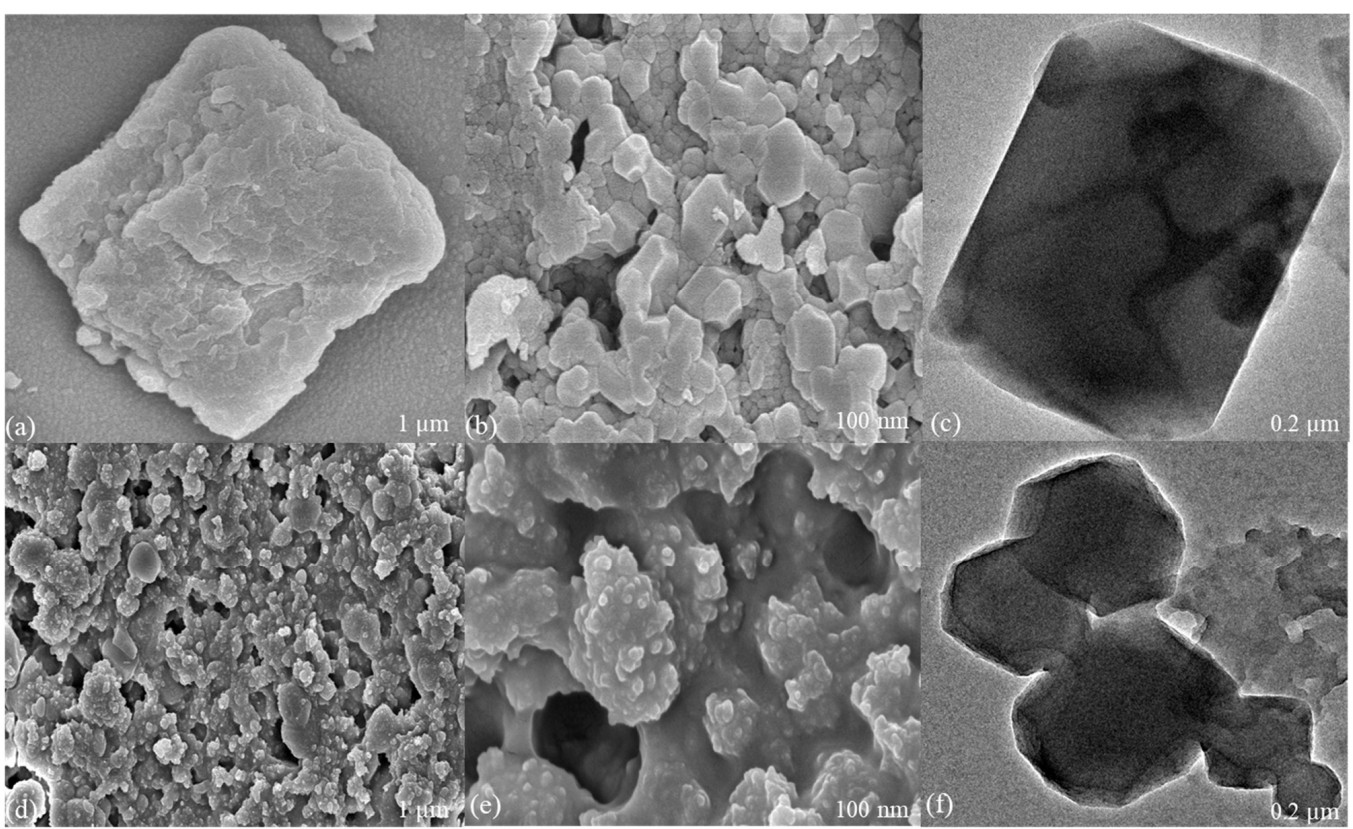

**Figure 5.** SEM and TEM images of $Cu_{10}ZIF-8$ (**a–c**) and $Cu_{20}ZIF-67$ (**d–f**) obtained by the IST approach.

**Table 3.** The optimum parameters for the reaction of both catalysts for CuAAC.

| Catalyst | Quantity | Time | Temperature | Solvent | Reusability | Substrate |
|---|---|---|---|---|---|---|
| $Cu_{10}ZIF-8$ | 0.001 g | 5 h | 85 °C | water | 4 cycles | $C_8H_6$, $C_7H_7Cl$ |
| $Cu_{20}ZIF-67$ | 0.001 g | 2 h | 90 °C | water | 4 cycles | $C_8H_6$, $C_7H_7Cl$ |

To investigate the recovery and reusability of the catalysts, after each cycle, both $Cu_{10}ZIF-8$ and $Cu_{20}ZIF-67$ were collected by centrifugation (5000 min, 5 min) after washing three times with methanol. After this, the catalysts were dried at 60 °C and reused for the next run without extra treatment. As shown in Figure 6, after three cycles, there was a slight decrease in yield. The loss of small amounts of catalysts after washing and the separation or agglomeration of the ZIFs during the reaction is most likely the main reason for the drop in catalytic performance. Additionally, to determine the crystallinity and morphology of the heterogeneous catalysts after the 4th cycle, SEM analysis was performed. The SEM image of the recovered nanocatalyst was similar to the fresh one, as no variation in the morphology could be observed. This also demonstrates the high structural stability possessed by $Cu_{10}ZIF-8$ and $Cu_{20}ZIF-67$. It is remarkable to indicate that the doping of copper did not affect the zeolitic imidazole framework's morphology.

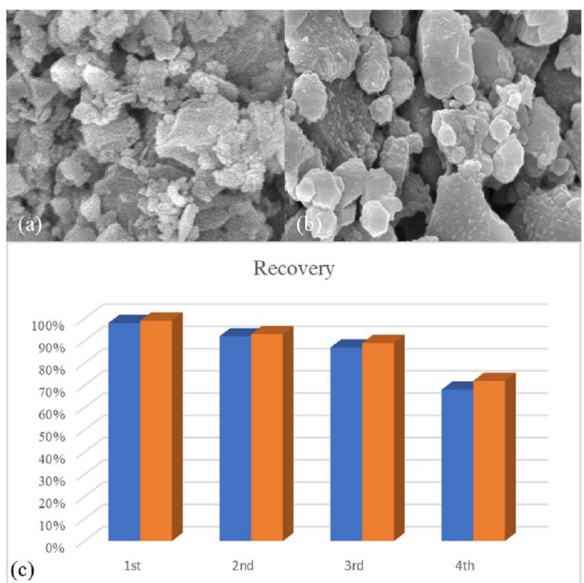

**Figure 6.** SEM images of $Cu_{10}ZIF-8$ (**a**) and $Cu_{20}ZIF-67$ (**b**) obtained by the IST approach after the 4th cycle; (**c**) obtained yields for four consecutive cycles by $Cu_{10}ZIF-8$ (blue) and $Cu_{20}ZIF-67$ (orange).

## 3. Discussion

According to experimental outcomes, a mechanism was proposed for the in situ thermal (IST) synthesis of bimetallic ZIFs. Temperature differences between the boiling and melting points of 2-MIM played a major role. The temperature used during the IST procedure was 200 °C. At this temperature, 2-MIM is molten and functions as a ligand for coordination with metal ions, acting as a hydrophilic agent affecting the as-formed framework structure and as a solvent for the reaction. The struggle among template-assisted and solvent exchanges was negligible, as only metal and ligand precursors were present in the reaction mixture during the IST process. Due to the stronger donor ability of the nitrogen in 2-MIM, the acetylacetonate in the metal acetylacetonate would be substituted by 2-MIM—thus generating a framework. Throughout the thermal process, the excess of 2-MIM could be easily removed, as its boiling point is substantially lower than 200 °C. Consequently, no further treatment or activation process was required to obtain highly pure ZIFs via the IST approach. Figure 7 displays the color changes in the ZIF-8 and ZIF-67 containing different percentages of copper loadings, obtained through the IST method.

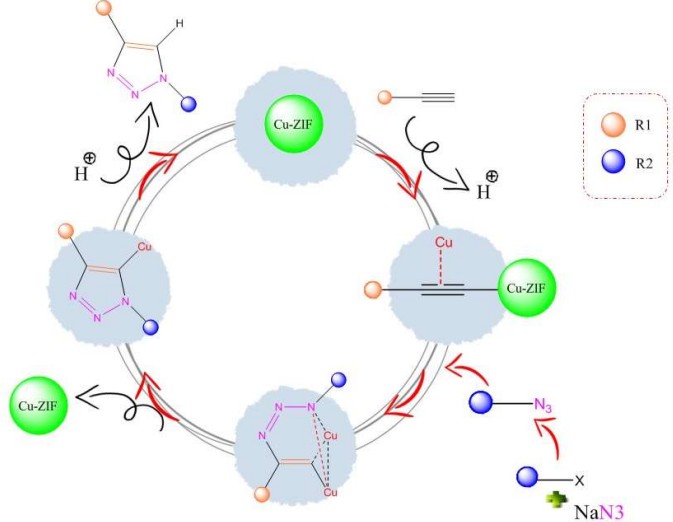

**Figure 7.** Demonstration of the mechanism and role of $Cu_{10}ZIF-8$ and $Cu_{20}ZIF-67$ catalysts in CuAAC.

## 4. Materials and Methods

### 4.1. Materials

Cobalt (II) acetylacetonate ($\geq$99%, Aldrich, Seoul, ROK), zinc acetylacetonate hydrate (99%, Aldrich), copper (II) acetylacetonate ($\geq$99.9%, Aldrich), 2-methylimidazole (99%, Aldrich), phenylacetylene (98%, Aldrich), benzyl chloride (99%, Aldrich), sodium azide (99%, Aldrich), and ethyl acetate (99%, Aldrich) were used as received.

### 4.2. Synthesis of Cu-ZIF-8 and Cu-ZIF-67

To synthesize Cu/ZIF-67, 2-methylimidazole (3 mmol, 246 mg) was mixed with cobalt (II) acetylacetonate (0.8 mmol, 205 mg) and copper (II) acetylacetonate (0.2 mmol, 52.2 mg). Then, the mixture was physically mixed in a mortar for 5 min under an ambient atmosphere.

The same procedure was applied to synthesize Cu/ZIF-8, except that the Co-source was replaced by the Zn-source; 2-methylimidazole (3 mmol, 246 mg) was mixed with zinc acetylacetonate hydrate (0.5 mmol, 131 mg) and copper (II) acetylacetonate (0.5 mmol, 104 mg). Then, the mixture was physically mixed in a mortar for 5 min under an ambient atmosphere.

Thereafter, the powder mixture was transferred in an alumina boat ($20 \times 100 \times 13$ mm$^3$) and placed in an alumina tube (OD: 60 mm, length: 1000 mm) inside the muffle furnace (SH 1500, SAMHEUNG Instrument Technology Co. Ltd., Sejong, Republic of Korea) under an inert atmosphere (N$_2$ stream (100 cm$^3 \cdot$min$^{-1}$)). A two-step temperature program was applied: from R.T. to 100 °C in 30 min, then from 100 °C to 200 °C (5 °C$\cdot$min$^{-1}$). Once the final temperature was reached, this temperature was held for one hour. After cooling, the final ZIFs were collected.

### 4.3. Characterization

The crystallinity of the obtained materials was determined by X-ray diffraction (XRD) using a Bruker D8 Advance diffractometer at 40 kV and 45 Ma, with a Cu K$\alpha$ radiation source ($\lambda$ = 1.54056 Å at 40 kV and 45 Ma) and a scanning rate of 2°/min$^{-1}$. The materials' size and morphology were determined by field-emission scanning electron microscopy (FE-SEM, Zeiss Ultra Plus, Potsdam, Germany). To determine the gas adsorption–desorption properties, an ASAP 2020 Analyzer (Micrometrics Instruments, Daejeon, Republic of Korea) was used to apply CH$_4$, CO$_2$, and N$_2$ gases of 99.999% purity. Additionally, samples were activated under a dynamic vacuum at 200 °C for 3 h before the adsorption measurements. The Brunauer–Emmett–Teller (BET) and Langmuir approaches were applied to measure the surface area and porosity. To investigate the thermal stability of the materials, thermogravimetric analysis (TGA) was performed on a Netzsch (STA449c/3/G) Instrument using a heating rate of 5 °C$\cdot$min$^{-1}$ under a nitrogen atmosphere; $^1$H NMR spectra were recorded on a Bruker Advance III 500 spectrometer in CDCl$_3$, and $^{13}$C NMR spectra were recorded in CDCl$_3$ on a Bruker Advance 500 (126 MHz) spectrometer. To determine the elemental composition, a VARIAN 720-ES Inductively Coupled Plasma-Optical Emission Spectrometer (ICP-OES) was used. Transmission electron microscopic (TEM) studies were performed on a Philips CM20 instrument operating at 200 kV. The samples were prepared by placing a drop of the ZIFs in ethanol onto a carbon film-supported copper grid. Functional group analysis of the materials was executed via Fourier transform infrared (FTIR) by applying a thermo Nicolet Avatar 360, USA, and KBr pellets.

### 4.4. Catalytic Reaction

The Azide-Alkyne cycloaddition reaction, see Scheme 1, was performed using Cu/ZIF-8 or Cu/ZIF-67 catalysts (1 mol% of catalyst) in water (2 mL) in a round bottom flask connected to a reflux condenser.

**Scheme 1.** Azide-Alkyne cycloaddition model reaction.

The temperature was set to 70 °C under argon. Sodium azide, phenylacetylene, and benzyl chloride were used as model substrates and were added to the catalyst and stirred for 3 h. Then, the catalyst was separated by centrifugation, washed with ethanol, and dried for the next run. The obtained product was purified by flash chromatography for $^1$H and $^{13}$C NMR analysis or gravimetric analysis. It is worth noting that numerous Cu sources have been documented as exhibiting selective catalytic activity in the CuAAC (Copper-Catalyzed Azide-Alkyne Cycloaddition) reaction. This transformation facilitates the synthesis of 1,4-disubstituted 1,2,3-triazoles from azides and terminal alkynes under ambient or mild conditions. In our study, we sought to compare the catalytic performance of our catalysts with other Cu-based catalysts previously reported in the literature. We observed superior reaction conditions, including enhanced reaction rates, higher conversion yields, and improved selectivity. Additionally, our catalysts demonstrated broader substrate compatibility, efficiency, and enhanced stability—surpassing those exhibited by other Cu sources in the CuAAC reaction.

*4.5. CuAAC Mechanism Using Cu/ZIF-8 or Cu/ZIF-67 Catalysts*

Sharpless and his peers came up with a process for the synthesis of 1,4-disubstituted 1,2,3-triazoles that involves the use of a copper alkynyl intermediate. Researchers have reported considerable enhancements in yield along with improvements in the reaction kinetics, in addition to improvements in regioselectivity. Cu in the oxidation state +1 may be produced in situ from $CuSO_4.5H_2O$ and sodium ascorbate, or it can be utilized directly to produce compounds such as CuI, $(Cu(PPh_3)_3Br)$, CuBr, and $Cu_2O$. Sharpless and his colleagues first claimed this was a mononuclear process; however, as scientific understanding developed, Finn and Folkin revealed a binuclear mechanism. A complex is formed when two copper atoms are bound to the alkyne in separate places. This complex is then attacked by the nucleophilic azide, which results in the formation of a ring with six members. Following this, the development of a new C–N bond ultimately resulted in the product that was intended, as seen in Figure 7 [15,16].

**5. Conclusions**

An eco-friendly, simple, and efficient in situ thermal approach (IST) was applied to synthesize zeolitic imidazole frameworks containing copper that have high porosity (Cu/ZIF-8 and Cu/ZIF-67). During the IST method, no solvent or additive was used to generate a high yield in a short time. Therefore, this method has great potential for being used for large-scale synthesis, as long as it is cost-effective. It was found that 2-methylimidazole has three important, simultaneous roles during IST synthesis: it acts as a solvent, as a hydrophilic agent, and as a ligand. The thermal process removes all side products or residues, if any, and forms a crystalline network. Hence, no further activation or purification steps are required. The synthesized copper-containing ZIFs were used as efficient catalysts for the Azide-Alkyne cycloaddition. Both Catalysts ($Cu_{10}ZIF-8$ and $Cu_{20}ZIF-67$) exhibited high catalytic performance in water as a green solvent. Finally, the IST approach could be a prominent candidate for the synthesis of pure metal-organic frameworks in a green fashion on a large scale.

**Supplementary Materials:** The following supporting information can be downloaded at: https://www.mdpi.com/article/10.3390/catal13061003/s1, Figure S1: IR pattern for $Cu_{10\%}$-ZIF-8; Figure S2: IR pattern for $Cu_{20\%}$-ZIF-67; Figure S3: $^1H$ NMR of 1-benzyl-4-phenyl-1H-1,2,3-triazole (product) from reaction catalyst by $Cu_{10\%}$-ZIF-8; Figure S4: $^{13}C$ NMR of 1-benzyl-4-phenyl-1H-1,2,3-triazole (product) from reaction catalyst by $Cu_{10\%}$-ZIF-8; Figure S5: $^1H$ NMR of 1-benzyl-4-phenyl-1H-1,2,3-triazole (product)from reaction catalyst by $Cu_{20\%}$-ZIF-67; Figure S6: $^{13}C$ NMR of 1-benzyl-4-phenyl-1H-1,2,3-triazole (product) from reaction catalyst by $Cu_{10\%}$-ZIF-67; Table S1: Applying different types of Organic halides and Azides to achieve 1,2,3 triazoles and its derivatives; Figure S7: Color changes with different percentages of Cu: (a) ZIF-67 (b) $Cu_{10}$ZIF-67 (c) $Cu_{20}$ZIF-67 (d) $Cu_{30}$ZIF-67 (e) $Cu_{40}$ZIF-67 (f) $Cu_{50}$ZIF-67 (g) ZIF-8 (h) $Cu_{10}$ZIF-8 (i) $Cu_{20}$ZIF-8 (j) $Cu_{30}$ZIF-8 (k) $Cu_{40}$ZIF-8, and (l) $Cu_{50}$ZIF-8; Figure S8: N2 adsorption/desorption curves for Cu-ZIF-8 and Cu-ZIF-67 materials with different amount of copper: (b) $Cu_{20}$ZIF-8, (c) $Cu_{30}$ZIF-8, (d) $Cu_{40}$ZIF-8, (e) $Cu_{50}$ZIF-8, (f) $Cu_{10}$ZIF-67, (h) $Cu_{30}$ZIF-67, (i) $Cu_{40}$ZIF-67, and (j) $Cu_{50}$ZIF-67; Table S2: Porosity and surface area of $Cu_x$ZIF-8 and $Cu_x$ZIF-67 samples; Figure S9: Energy dispersive spectrometry (EDS) mapping of $Cu_{20}$ZIF-67 obtained by the IST approach; Figure S10: graphical abstract.

**Author Contributions:** A.P.A.: Conceptualization, methodology, investigation, formal analysis, data curation, writing—original draft, validation, visualization. S.R.D.: Methodology, data curation, writing—original draft, validation. S.Z.: Conceptualization, methodology, formal analysis. P.M.H.: Visualization. S.C.: Conceptualization, methodology, validation, visualization, S.Z.: Resources, project administration, Writing—review and editing, F.V.: Conceptualization, methodology, validation, Writing—review and editing, supervision, project administration, Funding acquisition. All authors have read and agreed to the published version of the manuscript.

**Funding:** This research received no external funding.

**Data Availability Statement:** The raw data and the processed data required to reproduce these findings are available to download from the Supplementary Information.

**Acknowledgments:** F.V. and S.C. are grateful for the financial support from Wuhan University of Technology.

**Conflicts of Interest:** The authors declare no conflict of interest.

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
