# Peer review of "Green Approach for Synthesizing Copper-Containing ZIFs as Efficient Catalysts for Click Chemistry"

_catalysts, doi:10.3390/catal13061003_

Round 1
Reviewer 1 Report
Upon some searching, it appears that Cu doped ZIF-8 has been reported elsewhere before, e.g.
Catal. Sci. Technol., 2015, 5, 1829-1839
New J. Chem., 2019, 43, 18702-18712
Catal. Sci. Technol., 2019, 9, 2673-2681
Journal of CO2 Utilization, Volume 48, June 2021, 101523
Some of these references have very high Cu loadings in ZIF as well, so instead of addressing high Cu loadings, it will be better to focus on the novelty of this work of using a green approach during the synthesis.
For ZIFs, the stoichiometry between the metal and the linker is 1:2, authors were using a 1:3 ratio. I was wondering how the excessive linker was removed after the synthesis.
Minor editing of English language required
Reviewer 2 Report
Recommendation: Major revision
The study presents a convenient and efficient technique for investigating the catalytic performance of Azide-Alkyne cycloaddition in water, utilizing bimetallic-Cu-ZIF-8 and bimetallic-Cu-ZIF-67-based composite materials. While the article shows promise and warrants consideration for publication, several revisions are necessary to address the following issues:
1. Overall, the English in the article is good. However, some minor corrections are needed on pages 1 and 2. For example, the authors should use "a better" instead of "much better" and change “use in applications such as catalysis, biofuel production, drug delivery, and separation, as demonstrated by many reports” instead of “use for some applications such as...”
2. Please ensure that there is a space before citations throughout the manuscript.
3. It is important to accurately and correctly represent superscripts and subscripts in the manuscript.
4. Including chemical structures along with assigned peaks in the H1 and C13 Supplementary Information (SI) would be beneficial for readers.
5. The authors claim the utilization of a 17 moiety for Click reactions; however, in the SI, only the NMR data of a single moiety is mentioned. It would be helpful to provide more comprehensive information regarding the utilization of the 17 moieties.
6. Can the authors provide a demonstration of the mechanism and role of Cu10ZIF-8 and Cu20ZIF-67 catalysts in click reactions?
7. It is recommended to compare the catalytic capabilities of the catalysts with similar systems from the existing literature.
Overall, the English in the article is good. However, some minor corrections are needed on pages 1 and 2. For example, the authors should use "a better" instead of "much better" and change “use in applications such as catalysis, biofuel production, drug delivery, and separation, as demonstrated by many reports” instead of “use for some applications such as...”
Round 2
Reviewer 2 Report
The authors have done all the suggested changes, hence the present form of the manuscript can be accepted for publication.